# Which Seed Properties Determine the Preferences of Carabid Beetle Seed Predators?

**DOI:** 10.3390/insects11110757

**Published:** 2020-11-04

**Authors:** Hana Foffová, Sanja Ćavar Zeljković, Alois Honěk, Zdenka Martinková, Petr Tarkowski, Pavel Saska

**Affiliations:** 1Crop Research Institute, Functional Diversity in Agro-Ecosystems, Drnovská 507, Ruzyně, 161 06 Praha 6, Czech Republic; honek@vurv.cz (A.H.); martinkova@vurv.cz (Z.M.); saska@vurv.cz (P.S.); 2Department of Ecology, Faculty of Environmental Sciences, Czech University of Life Sciences, Kamýcká 129, Suchdol, 165 00 Praha, Czech Republic; 3Centre of the Region Haná for Biotechnological and Agricultural Research, Department of Genetic Resources for Vegetables, Medicinal and Special Plants, Crop Research Institute, Šlechtitelů 29, 783 71 Olomouc, Czech Republic; sanjacavarsc@gmail.com (S.Ć.Z.); Tarkowski@vurv.cz (P.T.); 4Centre of Region Haná for Biotechnological and Agricultural Research, Department of Phytochemistry, Palacky University, Šlechtitelů 27, 783 71 Olomouc, Czech Republic

**Keywords:** preference, ground beetles, weed seeds, seed properties

## Abstract

**Simple Summary:**

The carabid beetles are well known for the consumption of weed seeds in arable land, but how they choose the seeds is poorly known. In this work, we try to explain the patterns in preferences of 37 species of carabids based on eight seed properties of 28 species of seeds. Surprisingly, chemical properties of the seeds did not affect the preferences. Instead, preferences were driven mainly by seed structural properties. The importance of particular seed properties was also affected by the degree of predator specialization.

**Abstract:**

Ground beetles are important invertebrate seed predators in temperate agro-ecosystems. However, there is a lack of information regarding which seed properties are important to carabids when they select seeds for consumption. Therefore, seed properties, such as size, shape, morphological defence, and chemical composition, were measured, and in addition to seed taxonomy and ecology, these data were used to explain carabid preferences. Carabid preferences were assessed using a multi-choice experiment with 28 species of weed seeds presented to 37 species of Carabidae. Multiple regression on distance matrices (MRM) was used to determine the importance of particular sets of seed properties for carabids. The analysis was conducted for the full set of carabids (37 species) as well as for subsets of species belonging to the tribes of Harpalini or Zabrini. For the complete set of species, seed dimensions, seed mass, taxonomy, plant strategy, and seed coat properties significantly explained carabid preferences (proportion of explained variance, R^2^ = 0.465). The model for Harpalini fit the data comparably well (R^2^ = 0.477), and seed dimensions, seed mass and seed coat properties were significant. In comparison to that for Harpalini, the model for Zabrini had much lower explanatory power (R^2^ = 0.248), and the properties that significantly affected the preferences were seed dimensions, seed mass, taxonomy, plant strategy, and seed coat properties. This result suggests that the seed traits that carabids respond to may be specific to taxonomic and likely relate to the degree of specialisation for seeds. This study contributes to understanding the mechanisms that determine the preferences of carabid beetles for seeds.

## 1. Introduction

Ground beetles (Coleoptera: Carabidae) are among the most important groups of weed seed predators in temperate agro-ecosystems where they help to reduce weed seeds. These granivorous species of arable land mainly belong to the tribes Harpalini (e.g., genera *Harpalus*, *Ophonus*, *Acupalpus*, *Stenolophus,* or *Anisodactylus*) and Zabrini (genera *Amara* and *Zabrus*) [1,2], but species from other groups consume seeds as well. Recent findings suggest that granivory is more widespread within this family than previously thought [3,4]. Species that are specialised seed feeders often show distinct seed preferences [5]. Species of Zabrini prefer seeds of Asteraceae, Brassicaceae, or Caryophyllaceae, and they seem to be more selective than species of the tribe Harpalini, which prefer seeds of Violaceae or Asteraceae [6,7]. Many species of these families are considered to be problematic weeds. However, the knowledge on what drives the preferences is not fully understood. Predator identity, taxonomy, body size, size of mandibles [8,9], seed size, and other seed properties may affect carabid preferences for seeds. Understanding the driving factors of the preferences would potentially improve our ability to predict which seeds are the most vulnerable to which predators.

Seeds are usually unevenly scattered on the ground or aggregated in patches near the mother plant; therefore, insect seed predators have to locate the seeds or patches of seeds. However, the seeds try to resist predators. The defence of seeds against predators is divided into two main groups, morphological and chemical traits, which inevitably interact with each other and influence seed dormancy and persistence in soil [10,11], and in this way influence predation in the long term [12].

Although the information on the cues seed predators use is scarce and we hypothesized that the process that ultimately leads to seed predation is similar to the one described for other types of predators [13,14,15]. The typical process of prey location by an insect predator usually includes several steps, each having typical sets of cues involved. Visual or olfactory cues may be important when searching for seeds [16,17,18,19]. Utilising (semio-)chemicals is a common means of communication within food webs [20,21,22]. How important it is for seed predation is poorly understood. Few studies have shown that ground beetles detect volatiles from other animals, such as aphids, springtails, or slugs [23,24], as well as from plants [25,26] and probably seeds [16,17,24]. The chemical properties of seeds may change the behaviour of seed predators (serve as attractants or as repellents). The detection rate may be affected by the properties of seed coats because some are impermeable to gases, chemical compounds, or water [27]. This rate can also be affected by the level of imbibition [16] because the imbibed seeds release different amounts of volatile compounds, including carbon dioxide, alcohols, aldehydes, alkane, ketones, volatile acids [28], or ethylene [29], which can potentially attract or repel beetles.

Seeds vary in their morphological properties, such as mass, size and shape; as well as defensive structural traits. These properties affect seed interception and handling by a predator. Seed mass [30,31,32] and size [6,33,34,35] are major drivers in seed selection by ground beetles and there is a relationships between carabid body size and seed size or mass [6,7]. Larger seeds might be more apparent to predators [36], and they also stay on the soil surface for longer than smaller ones [37,38]. Seed shapes can also affect predation but has never been studied. The smaller, round seeds are able to escape seed predation more than flattened ones [31,39,40]. Round seeds fit better in cracks in the soil where they can escape predation [41]. In comparison to flat seeds, round ones can also be harder to handle because they pop out of mandibles (e.g., seeds of Amaranthaceae) [6].

Once seeds are found, a predator is expected to evaluate seed attractiveness. The chemical profile of the seed surface is often important in this process [17,22]. Waxes or fatty acids present on the seed surface [42] may drive a predator’s decision to feed on it or not [43,44]. Other surface compounds could also affect seed predation. Other surface compounds contain mostly long-chain alkanes or their branched counterparts, which are common constituents of plant waxes [45]. These compounds protect seeds from physical, temperature-related, or water damage to ensure that the plant seed remains in a state of dormancy [46]. Once a predator attempts in feeding, crushing and opening a seed is further affected by physical traits, such as thickness [5] or the strength of the seed coat [5,47]. These are seemingly related, but this is not necessarily the case (for example, seed coats can be relatively thin but hard, e.g., *Silene latifolia alba* (Mill.) Greut. et Burdet, or thick but soft, e.g., *Fumaria officinalis* L.). This type of physical defence is potentially more effective, and in comparison to other types of defences, less costly for the plant [48]. A higher investment in a seed coat may increase post-dispersal survival [10,11,48]. That seed coat thickness can be an adaptive defensive trait is supported by the finding of Benkman [49], who documented stronger seed coats in environments with predators rather than without predators. There is also a positive relationship between seed mass and seed coat thickness [50] as well as the interaction among seed size, mass, and strength of seed coat [5]. There may also be an interaction between seed coat hardness and shape, which may explain seed preference [9].

After successfully opening, a predator further evaluates the nutritional composition of a seed (amount of starches, proteins, oils, secondary metabolites, fatty acids, etc.) [22,42,51,52,53,54,55], which stimulates or deters the predator from additionally feeding on conspecific seeds. Some of the chemical compounds can be distasteful or poisonous for their predator (e.g., opium and L-dopa) [56], but insects have evolved systems to detoxify these compounds. In fact, we know only very little about this hypothetical sequence of events leading to the destruction of a seed by the mandibles of insect seed predators.

In addition to seed chemical and morphological properties, plant taxonomy [6] and the life cycle strategy of plants [57] are important determinants of predator preferences. The sister taxa of plants may be more attractive for seed predators than taxa unrelated ones [6], likely because related seeds have similar properties.

The aim of this study is to explore which weed seed properties are decisive for preferences of carabid beetles. We focus on properties related to seed size, shape, mass, and morphological defence; seed chemical properties (volatile compounds, fatty acids, and other surface compounds); and seed ecology and taxonomy.

## 2. Materials and Methods

### 2.1. Seed Material

A set of 28 species of weed seeds was used (Table 1). Each year, the seeds were hand-collected de novo from the parental plant at full maturity using laboratory gloves. The seeds were cleaned from dust and admixtures of non-seed plant particles by blowing, dried at room temperature (25 °C for 30 days) and then stored in the freezer (−21 °C) until the experiments.

### 2.2. Preference Experiments

The preferences of 37 species of carabids were evaluated (Table 1). The preferences were determined based on a cafeteria experiment described in Honěk et al. [6] and Saska et al. [7]. The seeds of 28 species of weeds (Table 1) were presented on tin trays filled with white modelling clay (JOVI, Barcelona, Spain). The seed trays were arranged in two concentric rings in Petri dishes (20 cm in diameter) with 10 beetles for five days. The seed consumption was counted daily, after which it was summed and standardised to remove the effect of carabid body size on the total consumption and be able to compare data across the species [6]. Standardisation was performed by converting the actual consumption of seed to the proportion of the most consumed seed.

### 2.3. Measurement of Seed Morphological Traits

Seed mass was measured by weighing 100 seeds on a balance to a precision of 10^−5^ g (Sartorius, Göttingen, Germany). Seed dimensions were measured following Bekker et al. [38], using a digital calliper and five seeds per species: A—the longest dimension, B—the longest dimension perpendicular to A within the same plane, and C—the longest dimension perpendicular to the plane of A and B.

These dimensions were used to calculate indices that describe seed shape, flatness, eccentricity, and volume. The shape of the seed was calculated as in Bekker et al. [38], $Vs=\;\sum^{\;}\frac{{\left(x\;-\;\bar{x}\right)}^{2}}{n}$, where *x* represents a division of either *A*, *B*, or *C* through *A* and $\bar{x}$ as their mean, and *n* is 3. vs. ranges from 0 for perfectly spherical seeds to 0.2 for seeds shaped like a thin disc or spindles. The flatness of the seeds [59] was calculated as $FI=\frac{\left(A\;+\;B\right)}{2\;*\;C}$. *FI* ranges from 1 for a complete sphere to higher values for plane-like or spindle-like seeds. The eccentricity of the seeds [59] was calculated as $EI=\;\frac{A}{B}$. *EI* ranges from 1 for round seeds to values > 2 for spindle-like seeds. The volume of the seeds was calculated as V=A∗B∗C [59].

Seed coat thickness was measured using a light microscope on sections of seeds. Dry seeds were infiltrated with a 2% sucrose solution for six hours at room temperature, mounted onto cryo-gel on the alum chuck, and sectioned using a cryotome (Shandon SME, Astmoor, UK). Sections were observed using an Olympus BX51 microscope (Olympus Corp., Tokyo, Japan) and documented with an Apogee U4000 digital camera (Apogee Imaging Systems, Inc., Roseville, CA, USA). Five seeds of each species were measured 10 times. The strength of the seed coat was measured on an MTS 02 (Aviko Praha, Praha, Czech Republic), which measures the force developed by the instrument to crack the seed coat [N]. For each species, 10 seeds were measured.

### 2.4. Chemical Analysis of Seeds

Seeds were subjected to detailed chemical analysis, which differed in the targeted groups of compounds and methods used to detect them. The targeted groups of compounds were considered to be perceived by carabids either from a distance or during handling seeds and included surface waxes, amino acids, and volatile compounds.

Fatty acids from the ground seeds (total fatty acids) as well as from seed surfaces were isolated and derivatized into corresponding volatile methyl esters and then quantified via gas chromatography–mass spectrometry (GC–MS) [60,61,62]. The isolation protocol was optimised for a small-scale experiment using ~50 mg of seeds for surface fatty acids and ~25 mg of seeds for total fatty acids. After isolation with a chloroform:methanol (2:1) mixture, the fatty acids were trans-esterified with a sodium methoxide solution into corresponding methyl esters and then extracted into *n*-hexane. The solvent was then removed under reduced pressure. Dry samples were dissolved in *n*-hexane containing 0.1% *n*-undecane as an internal standard for normalization of chromatographic conditions. All samples were analysed in triplicates. Identification and quantification of fatty acid methyl esters (FAMEs) in seed samples was accomplished via an internal standard calibration curve for 35 FAMEs (Supelco, Darmstadt, Germany). The single ion monitoring (SIM) mode was used for identification and quantification of each particular analyte.

Other surface compounds (waxes, alkanes, phytosterols, etc.) were isolated by dipping 50 mg of intact seeds into chloroform for 30 s [45]. After filtration, chloroform was evaporated under reduced pressure, and isolated compounds were dissolved into *n*-hexane containing 0.1% *n*-undecane as the internal standard for normalization of chromatographic conditions. Each seed sample was analysed in triplicate, and the results are presented as the percentage content of chloroform soluble surface compounds.

The volatile compounds from plant seeds were isolated and detected by the static headspace technique. For the analysis, sets of dry and imbibed seeds were used. Imbibed seeds produce other chemical compounds because of the start of the chemical processes during germination. The dry seeds were stored in the freezer. The imbibed seeds (0.5 g of each species) were incubated for 24 h at 25 °C before measurement. Volatile compounds released by seeds were pre-concentrated during incubation into headspace vials, and, therefore, we were able to detect them via a common GC–MS platform.

### 2.5. Ecology and Taxonomy of Plants

Information on plant ecology (annual, biennial, annual-biennial and perennial) and taxonomic placement were determined from the literature [58,63].

### 2.6. Data Analyses

The multiple regression on distance matrices (MRM) approach (ecodist package [64] for R version 3.4.1 [65]) was used for data analysis. MRM was preferred over other methods because it allows the regression of a response matrix on multiple explanatory matrices [66]. Raw matrices were created according to the nature of the data and possible correlation between the variables, presumed mechanisms behind the expected influence on the preferences, and methods used to generate them. Most of the available data were formed as two-dimensional matrices with seed species as rows and measured quantities as columns. We considered the following matrices for the initial exploration: carabid preferences (response matrix), seed mass (mass of 100 seeds in grams), seed dimensions (dimensions of the seed on axes A, B, and C), seed shape (indices of seed shape, flatness, eccentricity and volume), seed coat (seed coat thickness and strength), plant taxonomy (family of plants), plant strategy, volatile compounds from dry seeds, volatile compounds from imbibed seeds, fatty acids from seed surface, total fatty acids, and other surface compounds. Before the MRM approach can be applied, raw data matrices must be converted into distance matrices using the vegan package [67]. Bray–Curtis dissimilarities were used to convert the seed preference matrix because consumption was standardised on a scale of 1 to 0. The Mahalanobis distance was used for matrices of seed dimensions, seed coat, surface fatty acids, total fatty acids, volatile compounds from dry seed, volatile compounds from imbibed seeds and other surface compounds because these factors contain continuous numerical variables. The data matrices of seed mass, taxonomy and plant ecology were transformed to distance matrices by using the specified distance measurement. Prior to the analysis, the correlation between the dissimilarity matrices was explored by using Mantel’s permutation test for similarity of two matrices (999 permutations). The following matrices were excluded since they showed correlation with other matrices: seed shape (with seed dimensions; *p* > 0.001), volatile compounds from dry seeds (with volatile compounds from imbibed seeds; *p* > 0.009), and fatty acids from seed surface (with other surface compounds and taxonomy; *p* > 0.011). The distance matrix for carabid preferences was used as a response, and the following distance matrices were used as explanatory terms: seed dimensions, seed mass, seed coat, taxonomy, plant strategy, total fatty acids, other surface compounds and volatile compounds from imbibed seeds. Three different models were fitted that differed according to the carabid preference distance matrix: (i) one model was based on the full set of 37 carabid species, (ii) another model was calculated only for the species of Harpalini (15 species), and (iii) the final model was calculated only for the species of Zabrini (18 species). The variances with associated *p*-values from the multiple regressions were obtained using Legendre et al. (1994)’s permutation test with 9999 permutations [68]. The level of significance to reject the null hypothesis was set to α < 0.1.

## 3. Results

### 3.1. Preferences of Carabids

Seed consumption varied among the species of carabids [7]. The highest preferences by Harpalini were on seeds of *Cirsium arvense*, *Viola arvensis,* and *Cichorium intybus*, while tribe Zabrini preferred seeds of *Taraxacum officinale*, *Tripleurospermum inodorum,* or *Crepis biennis.* The small seeds of Brassicaceae were preferred by small carabids of both tribes. The standardized consumption of all species is in Appendix A.

### 3.2. Morphological Analysis of Seeds

The seed mass of 100 seeds ranged from 0.08 g (*Potentilla argentea* L.) to 8.72 g (*Arctium lappa* L.) (Appendix A). The seed dimensions were diverse and ranged from 9.076 ± 1.264 mm (dimension A of *Bidens tripartita* L.) to 0.272 ± 0.372 mm (dimension C of *Arenaria serpyllifolia* agg.). The shape index ranged from 0.171 ± 0.002 (*B. tripartita*) to 0.005 ± 0.003 (*Fumaria officinalis* L.). The flatness index ranged from 13.496 ± 0.73 (*B. tripartita* L.) to 1.144 ± 0.052 (*F. officinalis* L.) (Appendix A). Eccentricity ranged from 6.319 ± 0.322 (*Crepis biennis* L.) to 1.03 ± 0.012 (*Stellaria media* (L.) Vill.). The volume ranged from 16.932 ± 0.926 mm^3^ (*A. lappa*) to 0.058 ± 0.006 mm^3^ (*A. serpyllifolia*) (Appendix A). The strength of the seed coat varied among the species and families as well. The seeds of *Galium aparine* L required the greatest amount of power (99.47 ± 16.818 N) to crush the seed coat, and *Urtica dioica* L. required the least amount of power to crush the seed coat (1.14 ± 0.533 N). The seed coat thickness ranged from 0.138 ± 0.043 mm (*A. serpyllifolia*) to 0.017 ± 0.006 mm (*G. aparine*) (Appendix A).

### 3.3. Chemical Analyses of Seeds

The majority of the 35 fatty acids from the FAME standard mixture was detected in the analysed seeds (Appendix A). The greatest concentration of all fatty acids was found in the seeds of *Galinsoga parviflora* Cav. (467.75 ± 8.40 mg/g dry weight), while the lowest concentration was extracted from *G. aparine* (33.39 ± 1.26 mg/g DW). The major fatty acid in all seeds analysed was unsaturated linoleic acid, which accounted for ~50% of the total fatty acids quantified. The composition of the surface fatty acids varied between the species more than the composition of total fatty acids (Appendix A). The highest content of the sum of all surface fatty acids was found in the seeds of *Cirsium arvense* (114.12 ± 3.21 mg/g DW). Other species had a lower content of surface fatty acids. The lowest amount of all surface fatty acids (6.07 ± 0.32 mg/g DW) was found in *Amaranthus retroflexus* L. (Appendix A). Some of the fatty acids were found just in one species (e.g., cis-5, 8, 11, 14, 17-Eicosapentanoic acid in *Lapsana communis* L.).

Thirteen volatile compounds were detected in seeds (Appendix A). The amounts of volatile compounds varied between dry and imbibed seeds. The highest amount of volatile compounds was found in the seeds of *Sisymbrium loeselii* L. (4.8% of determined volatiles); while no volatile compounds were detected in *A. retroflexus* (Appendix A). The highest amounts of volatile compounds were found in seeds of *S. loeselii* (4.83% of determined volatiles), and the lowest amount was found in seeds of *A. retroflexus*, where 0.03% of volatiles were detected (Appendix A).

Nineteen other surface compounds (Appendix A) were detected in the seeds including long-chain alkanes or their branched counterparts, with significant amounts of phytosterols, such as β-sitosterol, were detected. The composition of the other surface compounds also varied between the species.

### 3.4. Relationships among Carabid Preferences and Seed Properties

The full model on standardized consumption included matrices on seed mass, seed dimensions, seed coat, seed taxonomy, plant strategy, other surface compounds, total fatty acids, and volatiles released from imbibed seeds. The model explained the variation in consumption across the range of carabid species (R^2^ = 0.465, *p* = 0.001; Figure 1a), with the following matrices contributing significantly (at the level of α = 0.1) to the explained variance: seed dimensions (*p* < 0.001), seed coat (*p* < 0.001), taxonomy (*p* = 0.035), seed mass (*p* = 0.054), and plant strategy (*p* = 0.058).

However, by re-running the analysis separately for the two major taxonomic groups of carabids, Zabrini and Harpalini, we found specific responses. The model for Harpalini fit the data comparably well to the global one (R^2^ = 0.477, *p* = 0.001; Figure 1b), with the following matrices contributing significantly: seed dimensions (*p* < 0.001), seed coat (*p* < 0.001) and seed mass (*p* = 0.062). In no model did seed phytochemistry significantly influence the seed preferences of the carabid beetles included in this study. The model for Zabrini had much lower explanatory power (R^2^ = 0.248, *p* = 0.001; Figure 1c), with the following matrices contributing significantly: seed coat (*p* = 0.002), seed dimensions (*p* = 0.005), taxonomy (*p* = 0.005), plant strategy (*p* = 0.036), and seed mass (*p* = 0.075).

## 4. Discussion

Seed properties, such as seed dimensions, mass, taxonomy, plant strategy, and physical seed coat traits were the most important properties affecting the preferences of carabid beetles in this study. The seed dimensions explaining over 13% of the preferences was the main factor affecting seed selection [6,33,34], even when other properties are considered. The interaction between seed size and the mass of carabids should not, however, be overlooked [6,7,53]. The size of the seeds also affects their chemical properties, such as the oil content or stored energy [55], which may affect seed predation.

The properties of the seed coat were also important because the seed coat protects seeds against predators. To open seeds, many species of carabids have evolved broad mandibles with large adductors [69] and bases that are generally triangular. The shape of mandibles varies among the tribes. Species of Harpalini have more asymmetrical mandibles than those of Zabrini. Quadrate mandibles with broadly rounded incisors and a basal face suggest an omnivorous diet in most Harpalini [70]. Species of the tribe Zabrini have short, square-shaped mandibles that are blunt at the tips, and are more adapted for crushing hard seeds [8]. This can explain why the seed coat properties were important properties for seed preference by Harpalini (13% of explained variance) but less so for Zabrini (5% of explained variance).

Seed preferences by species of the studied Zabrini species were related to seed taxonomy, which probably drove the significant response for the entire species set because for the tribe Harpalini, seed properties related to their taxonomy did not appear to be a significant determinant of preferences. This result is in line with the previous findings that Harpalini are less specialised than Zabrini [6]. Our results suggest that the seed traits to which the carabid seed predators respond may be species- or higher taxon-specific and perhaps dependent on the degree of carabid specialisation for seeds. Since Zabrini species are more specialised [6,71] to a narrower range of seeds, often with the same ecology or from the same family, the variables that appeared to be the most influential for preference determination were unexpected. On the other hand, Harpalini species (Figure 1b) are more generalists; therefore, it is ecologically sound that seed mass and dimensions would be the major variable explaining the variation among the matrices of traits for this tribe.

Seed chemistry did not seem to play a crucial role in seed selection by carabid beetles. Although other studies [16,24] have determined that volatiles originating from seeds can attract seed predators, our data do support these observations. There may be several reasons for this lack of support. The seeds used in the multi-choice experiment were dried and mounted on modelling clay [6], which could have limited the amount of volatiles released from the seeds [28,29] compared to those present on the soil surface. The other reason for this difference could be due the cold storage of seeds prior to seed preference assays. Although cold storage does not affect seed viability [72,73], defrosting could have potentially changed the qualitative and quantitative aspects of seed chemical ecology. This needs to be studied. The seeds could have been contaminated by fungi or bacteria [74], which also release their own suite of chemicals. In fact, ethanethiol that was found in the headspace of the tested seeds in our work suggests that some seeds were contaminated, most likely with methanogenic bacteria [75,76]. However, this occurrence should not be considered a problem because in the field, seeds are also colonised by microorganisms [77,78], so the interaction among seeds, microorganisms, and seed predators should be considered a natural component of seed predation and represents an interesting direction for future research.

## 5. Conclusions

Our data suggest that seed morphological properties are more important than chemical properties in determining the preferences of granivorous carabid beetles. Seed dimensions and seed coat properties were among the most important seed properties affecting carabid preferences. The preferences varied between the taxonomical groups of predators that differ in the degree of specialisation. This paper expands the knowledge on how seed defences influence seed preferences of carabid beetles.

## Figures and Tables

**Figure 1 insects-11-00757-f001:**
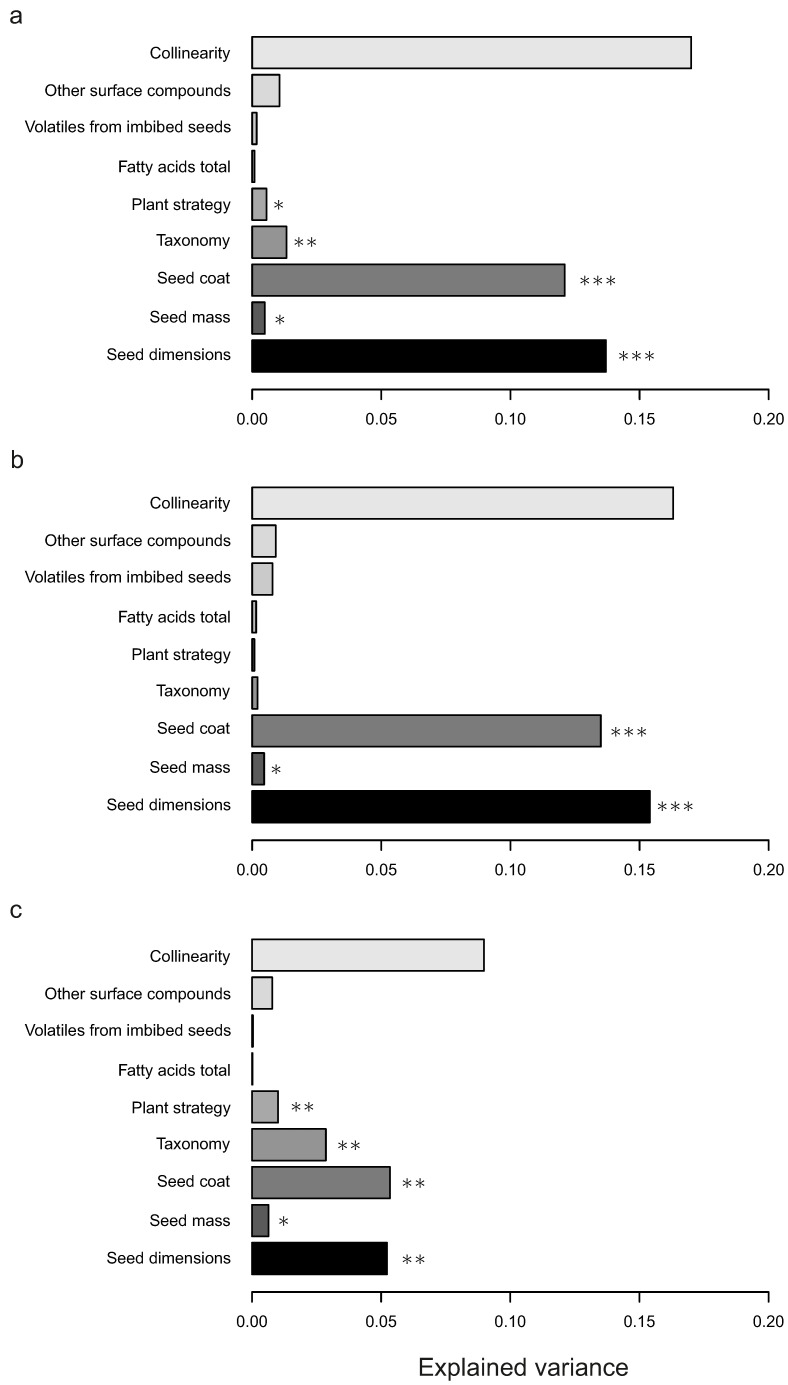
Contribution of the matrices of seed traits to seed preferences of carabid beetles based on a multiple regression on distance matrices (MRM) approach. The horizontal bars indicate the proportion of explained variance by a particular variable in the data. The collinearity shows part of the variation explained by the model, but which cannot be attributed solely to any of the single factors. (**a**) Full set of the 37 species of carabids (proportion of explained variance, R^2^ = 0.465, *p* = 0.01). (**b**) Species of the tribe Harpalini (R^2^ = 0.477, *p* = 0.001). (**c**) Species of the tribe Zabrini (R^2^ = 0.248, *p* = 0.001); * *p* < 0.1, ** *p* < 0.05, *** *p* < 0.001.

**Table 1 insects-11-00757-t001:** List of the model organisms that were used in the preference experiment. The plant taxonomy was based on Kubát et al. [58] while that of carabids on Hůrka [2].

Plants		Carabids	
Species	Family	Species	Tribe
*Amaranthus retroflexus* L.	Amaranthaceae	*Acupalpus meridianus* (Linnaeus)	Harpalini
*Arctium lappa* L.	Asteraceae	*Amara aenea* (DeGeer)	Zabrini
*Arenaria serpyllifolia* agg.	Caryophyllaceae	*Amara anthobia* (A. Villa et G.B. Villa)	Zabrini
*Bidens tripartita* L.	Asteraceae	*Amara apricaria* (Paykull)	Zabrini
*Campanula trachelium* L.	Campanulaceae	*Amara aulica* (Panzer)	Zabrini
*Capsella bursa-pastoris* (L.) Med.	Brassicaceae	*Amara bifrons* (Gyllenhal)	Zabrini
*Chenopodium album* L.	Amaranthaceae	*Amara consularis* (Duftschmid)	Zabrini
*Cichorium intybus* L.	Asteraceae	*Amara convexior* (Stephens)	Zabrini
*Cirsium arvense* (L.) Scop.	Asteraceae	*Amara convexiuscula* (Marsham)	Zabrini
*Consolida regalis* S.F. Gray	Ranunculaceae	*Amara eurynota* (Panzer)	Zabrini
*Crepis biennis* L.	Asteraceae	*Amara familiaris* (Duftschmid)	Zabrini
*Descurainia sophia* (L.) Prantl	Brassicaceae	*Amara ingenua* (Duftschmid)	Zabrini
*Fumaria officinalis* L.	Papaveraceae	*Amara litorea* (C.G.Thomson)	Zabrini
*Galinsoga parviflora* Cav.	Asteraceae	*Amara montivaga* (Sturm)	Zabrini
*Galium aparine* L.	Rubiaceae	*Amara ovata* (Fabricius)	Zabrini
*Lapsana communis* L.	Asteraceae	*Amara sabulosa* (Audient-Serville)	Zabrini
*Leonurus cardiaca* L.	Lamiaceae	*Amara similata* (Gyllenhal)	Zabrini
*Lepidium ruderale* L.	Brassicaceae	*Amara spreta* (Dejean)	Zabrini
*Melilotus albus* Med.	Fabaceae	*Anisodactylus signatus* (Panzer)	Harpalini
*Potentilla argentea* L.	Rosaceae	*Calathus ambiguus* (Paykull)	Sphodrini
*Silene latifolia alba* (Mill.) Greut. et Burdet	Caryophyllaceae	*Calathus fuscipes* (Goeze)	Sphodrini
*Sisymbrium loeselii* L.	Brassicaceae	*Harpalus affinis* (Schrank)	Harpalini
*Stellaria media* (L.) Vill.	Caryophyllaceae	*Harpalus atratus* (Latreille)	Harpalini
*Taraxacum officinale* agg.	Asteraceae	*Harpalus distinguendus* (Duftschmid)	Harpalini
*Thlaspi arvense* L.	Brassicaceae	*Harpalus honestus* (Duftschmid)	Harpalini
*Tripleurospermum inodorum* (L.) Schultz-Bip.	Asteraceae	*Harpalus luteicornis* (Duftschmid)	Harpalini
*Urtica dioica* L.	Urticaceae	*Harpalus rubripes* (Duftschmid)	Harpalini
*Viola arvensis* Murray	Violaceae	*Harpalus signaticornis* (Duftschmid)	Harpalini
		*Harpalus subcylindricus* (Dejean)	Harpalini
		*Ophonus azureus* (Fabricius)	Harpalini
		*Parophonus maculicornis* (Duftschmid)	Harpalini
		*Pseudoophonus griseus* (Panzer)	Harpalini
		*Pseudoophonus rufipes* (DeGeer)	Harpalini
		*Pterostichus melanarius* (Illiger)	Pterostichini
		*Stenolophus teutonus* (Schrank)	Harpalini
		*Trechus quadristriatus* (Schrank)	Trechini
		*Zabrus tenebrioides* (Goeze)	Zabrini

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
