# Peer review of "Which Seed Properties Determine the Preferences of Carabid Beetle Seed Predators?"

_insects, 2020, doi:10.3390/insects11110757_

Round 1

Reviewer 1 Report

Many carabids consume seeds but this aspect of their feeding ecology is understudied. The authors used laboratory tests and measured several weed seed traits in an attempt to identify relevant seed parameters that make them attractive to carabids. This is interesting work and a good research question. Overall, the design and presentatin is acceptable. Language use needs to be improved. I made several suggestins dierectly in the MS.

Swpecific comments:

L23 - not clear what 'full set of carabids' can mean? Rephrase

L24 what this R2 value means? Not clear - maybe you ought to move this elsewhere

L26 unclear what "data" fit the Harpalini - this is a little confusing. What is the response parameter, i.e. the 47% variation of what are explained by the model? This needs clarification.

L48-52 there is a jump in the argument here. First, you mention the location problem but do not bring that to any conclusion - then you jump to seed defensive traits and it is not clear why - or how these two relate to each other. Rephrase, and maybe extend.

L60 also from caterpillars: see Ferrante et al. BioControl, 2017

L119 what is the 'shortest possible time'? Indicate a range.

par starting on L139 - rearrange so that the equations are on their own lines, and the suymbol explanaitons are afterwards.

L149 how thick were the sections?

L161 write out what GC-MS is?

L264-5 add a little mroe text, explaining rather than pointing to the appendix. Avoid statements like the second sentence. mention tha min-max data, or if there were any species not consuming any seeds? Thsi shodul nto be longer than a few sentences.

In the results, the letter size of scientific names seems to be smaller. Modify.

L232 rephrase as 'The smallest seed was that of XX (mean=, SD=, n=) etc.

Fig 1 instead of a-b-c, label the panels directly with the categories. Use a dot plot for visualising the data. Do not put signficiance cods into the data rectangle - place them in the caption.

L288 but only physical coat traits, correct? If so, specify.

L308-9 this is an odd statement - taxonomy influences nothing - it is a theoretical construction. Perhaps better to turn it around: Seed properties were related to taxonomy.

Ref. item 71 - the correct name is Zetto Brandmayr, T

Author Response

Dear reviewer,

thank you so much for your comments and suggestions. They were helpful and improved the paper. We are also thankful for improving the English. Most of your suggestions were considered and included in the updated manuscript.

Specific comments:

Abstract:

The set of carabids was changed to the “full set of carabids (37 species)” L23-24.

The sentence with the R2 (L25-26) was better explained “For the complete set of species, seed dimensions, seed mass, taxonomy, plant strategy and seed coat properties significantly explained carabid preferences (proportion of explained variance, R2= 0.465).“

“L26 unclear what "data" fit the Harpalini” I hope that the previous change made the sentence clearer. 

Introduction:

L48-52 - The new sentence was added to improve the flow of the text to L48-52.  “However, the seeds try to resist predators. The defence of seeds against predators is divided into two main groups, morphological and chemical traits, which inevitably interact with each other and influence seed dormancy and persistence in soil [10,11], and in this way influence predation in the long term [12].”

The suggested paper (“L60 - Ferrante et al. BioControl, 2017”) was not included, because there was no information about volatile compounds which are coming from the caterpillars.

Material and Methods:

“L119 what is the 'shortest possible time'? Indicate a range.”   We decided to remove this sentence from the manuscript, because it varied between the species and all samples stayed it the freezer before the chemical analyses.

“L139 - rearrange so that the equations are on their own lines” - The equation was moved to the next line.

“L149 - how thick were the sections?“ The thickness of the sections varied between seeds based on other properties of the seeds and it do not matter on the measurement of the seed coat properties

“L161“ - The GC-MS was rewritten to Gas chromatography–mass spectrometry.

Results:

L264-5 Some more text was added to the preferences of the carabids. “Seed consumption varied among the species of carabids [7]. The highest preferences by Harpalini were on seeds of Cirsium arvense, Viola arvemsis and Cichorium intybus, while tribe Zabrini preferred seeds of Taraxacum officinale, Tripleurospermum inodorum or Crepis biennis. The small seeds of Brassicaceae were preferred by small carabids of both tribes. The standardized consumption of all species is in Appendix S1: Table 1.”

The scientific names were changed to the same size as the other.

We did not rephrase L232 as "The smallest seed" etc. There are multiple measurements related to size, so choosing one would be subjective. We prefer providing details for all the measurements. 

“Figure 1” was changed based on the suggestions of all the reviewers. We also changed the legend.

“L288 but only physical coat traits, correct?”Yes, it was just the physical coat traits – It was specified in the manuscript.

“L308-9 this is an odd statement“ L308-9 was changed to better meaning: Seed properties are related to their taxonomy.

Ref. item 71 was changed to the correct name: 71.  Zetto-Brandmayr, T. Spermophilus (seed-eating) ground beetles: first comparison of the diet and ecology of the harpaline general Harpalus and Ophonus (Col., Carabidae). In The Role of Ground Beetles in Ecological and Environmental Studies, Stork, N., Ed. Andover, 1990; pp. 307–316.

Thank you again, Hana Foffová

Reviewer 2 Report

Notes to Authors

Insects Manuscript: Which seed properties determine preferences of carabid beetle seed predators?

The authors describe an extensive study in which they characterize weed seeds and carabid beetle preference for these weed seeds. The authors have done a significant amount of work in the characterization of these seeds and the susceptibility of preference for the seeds. The manuscript provides a detailed examination of an ecologically important seed predator.

One overall detail that is missing from both the introduction and discussion is why does this matter? What is the benefit or detriment of carabids eating these seeds? If they are characterized as weed seeds, shouldn’t their eating them be beneficial to the environment, or is eating them creating a food web or part of a larger ecosystem process? Just to say that the carabids are the most important invertebrate seed predators doesn’t give me the context of why seed predators matter.

Specific Comments:

Supplemental Tables 3-7: You have a lot of n.d. listed. Why is that? Please briefly comment in the results.

Line 24: Belonging to Harpalini and/or Zabrini – please identify that these are tribes of Carabidae.

Intro: Why does the relationship between weed seeds and carabids matter? They are important seed predators but how does that play into a bigger picture? Just a sentence or two would be nice to explain how this impacts the ecosystem.

Lines 224-225: Just saying this and then pointing the reader to the appendix is not acceptable. Some very general overview statements are needed. Plus the table is extremely full of information and the reader will not take the time to sort through everything. General statements such as Group A preferred seeds from X more and Group B preferred seeds from Y more. And so on. Just 4-5 summarizing statements.

Line 227-237: Why are the species names so small?

Lines 232-233: Volume needs a label. I would also make the ranges go from smallest to largest. You do that with the first mass statement but not for the rest of the ranges.

Figure 1: Please explain what collinearity is in your figure, either in the caption or somewhere in the document text when you are describing the relationships.

Discussion: Again, why do these patterns matter? What does this suggest about food webs? Why should we track which Carabids are predators to which seeds? What do these data further inform on?

Line 327: Suit should be suite.

Author Response

Dear reviewer,

Thank you for your suggestions and comments on the manuscript. They were helpful and improved the paper.

“One overall detail that is missing from both the introduction and discussion is why does this matter? What is the benefit or detriment of carabids eating these seeds? If they are characterized as weed seeds, shouldn’t their eating them be beneficial to the environment, or is eating them creating a food web or part of a larger ecosystem process? Just to say that the carabids are the most important invertebrate seed predators doesn’t give me the context of why seed predators matter.”

The information why the preferences of carabids are important were added to introduction and discussion. The aim and the importance of this study were added to these parts.

Specific comments:                    

Abstract:

L24 – The word tribe was added

Introduction:

The introduction was already changed based on the previous comments.

Results:

“Lines 224-225: Just saying this and then pointing the reader to the appendix is not acceptable. Some very general overview statements are needed…. ” The preferences statements were changed and some more information about preferences of the carabids were attached: Some more text was added to the preferences of the carabids. “Seed consumption varied among the species of carabids [7]. The highest preferences by Harpalini were on seeds of Cirsium arvense, Viola arvemsis and Cichorium intybus, while tribe Zabrini preferred seeds of Taraxacum officinale, Tripleurospermum inodorum or Crepis biennis. The small seeds of Brassicaceae were preferred by small carabids of both tribes. The standardized consumption of all species is in Appendix S1: Table 1.”

L 227-237: The size of the Latin names was changed.

L 232-233: The volume labels were added.

“Figure 1” was changed based on the suggestions of all the reviewers. The legend was changed and the comments of the graphs were rewritten.

Discussion:

Line 327: Suit was change to suite.

“Supplemental Tables 3-7: You have a lot of n.d. listed. Why is that? Please briefly comment in the results.”   Supplemental Tables 3-7: In chemical analyses, many chemical compounds were not detected in the full set of the seeds. Comment of this was added to the results.

Thank you,

Hana Foffová

Reviewer 3 Report

The MS is clearly written and suitable for publication after some minor revision. The strength of the study is on the big amount of variables used. The authors as experts have a good knowledge of the field and conducted the study in the correct way. The article contains a good Introduction, but the description of the importance of the research should be more emphasized. In the conclusions you should treat the possible future development of your study, or what are the topics that need further researches, for example the relationships among carabid adaptation to a given environment and seed preference.

The paper could be improved by adding, if possible, a b/w picture of mandible shape of Harpalini and Zabrini.

At the end of the Introduction write clearly what is the aim of the paper.

Table 1 should be split into two sub-tables, i.e. Table 1a for plants and Table 1b for carabids.

Figure 1: grey values are difficult to follow, maybe you could cross-hatch some bars. In the legend it written “vertical bars”, probably they are horizontal bars.

320 - it is not clear if your data support or don’t support previous observations.

Author Response

Dear reviewer,

Thank you for your suggestions and comments on the manuscript.

More information about the importance of seed predators was added to the manuscript. The suggested figure of the mandibula of Harpalini and Zabrini were not added; we feel that this was not relevant for the major message of the paper.

The table was not changed because in the present form it saves place in the manuscript. The figure was changed based on the suggestion of all reviewers. "Vertical bars”, were change to horizontal bars.

Thank you,

Hana Foffová

Round 2

Reviewer 2 Report

Paper is much improved. Please proof read carefully for editorial and grammatical errors.